

# Vertical profiles of sub-micron aerosol single scattering albedo over Indian region immediately before monsoon onset and during its development: Research from the SWAAMI field campaign

Mohanan. R. Manoj[1], Sreedharan. K. Satheesh[1,2], Krishnaswamy. K. Moorthy[1], Hugh Coe[3]

[1]Centre for Atmospheric and Oceanic Sciences, Indian Institute of Science, Bangalore, India
[2]Divecha Centre for Climate Change, Indian Institute of Science, Bangalore, India
[3]Centre for Atmospheric Science, School of Earth and Environmental Sciences, University of Manchester, Manchester, UK

*Correspondence to*: M. R. Manoj (manojshibika@gmail.com)

**Abstract.** Vertical structures of aerosol single scattering albedo (SSA), from near the surface through the free troposphere,
have been estimated for the first time at distinct geographical locations over the Indian mainland and adjoining oceans, using in-situ measurements of aerosol scattering and absorption coefficients aboard the FAAM BAe-146 aircraft during the South West Asian Aerosol Monsoon Interactions (SWAAMI) campaign from June to July 2016. These are used to examine the spatial variation of SSA profiles and also to characterize its transformation from just prior to the onset of Indian Summer Monsoon (June 2016) to its active phase (July 2016). Very strong aerosol absorption, with SSA values as low as 0.7, persisted
in the lower altitudes (< 3 km) over the Indo-Gangetic Plains (IGP), prior to the onset of monsoon, with a west-to-east gradient; lower values occurred in the north-western arid regions, peaking in the central IGP and somewhat decreasing towards the eastern end. During the active phase of the monsoon, the SSA is found to increase remarkably, indicating far less absorption. Nevertheless, significant aerosol absorption persisted in the lower and middle troposphere over the IGP. Inputting these SSA and extinction profiles in a radiative transfer model, we examined the effects of using height-resolved information in estimating
atmospheric heating rates due to aerosols, over similar estimates made using a single columnar value. It was noticed that use of a single SSA value leads to an underestimation (over-estimation) of the heating rates over regions with low (high) SSA emphasizing the importance of height resolved information. Further, the use of realistic profiles showed the significant heating of the atmosphere by sub-micron aerosol absorption at the middle troposphere, which would have strong implications for clouds and climate.

## 1 Introduction

The Indian Summer Monsoon (ISM) is a large-scale climate system, having direct implications for South Asia. The role of atmospheric aerosols (both natural and anthropogenic) in significantly perturbing the ISM through direct and indirect radiative interactions have been well recognized (Ramanathan et al., 2001; Lau et al., 2006; Gautam et al., 2009; Jones et al., 2009; Vinoj et al., 2014; Solmon et al., 2015; Jin et al., 2016); yet remain not accurately quantified, especially due to the complex



nature of the interactions on the one hand and the highly complex aerosol system (being a mixture of abundant natural and anthropogenic species) that shows large heterogenity in space and time (for example Moorthy et al. (2016) and references there in). As such, aerosol-monsoon interaction over South Asia remains a very active topic in climate research (Kuhlmann and Quaas, 2010; Turner and Annamalai, 2012; Bollasina et al., 2014; Li et al., 2016; Nair et al., 2016). Improving the accuracy

of the description of aerosol perturbation of atmospheric thermodynamics requires critical information on aerosol properties. One of the most important of these are the spectral optical depth and size distribution, and the vertical variation of aerosol single scattering albedo (SSA), which is a measure of the relative contribution of scattering and absorption by aerosol to the total light extinction. Absorbing aerosol layers can critically modify the thermal structure of the atmosphere; hence information on both the vertical distribution of aerosols and type of aerosols at various altitudes are essential to understand their net

radiative impact. The relative position of clouds with respect to the aerosol layers also is important.

The importance of the information on vertically resolved aerosol properties in improving the accuracy of aerosol direct and indirect radiative forcing have been emphasized by a few regional studies. Clarke et al. (2004) have advocated the need to study the vertical heterogeneities of aerosol properties, as the values estimated for near-surface aerosols poorly represent the variations within the column. Using air-borne lidar measurements, Satheesh et al. (2008) have shown the role of elevated

aerosol layers over the Indian peninsula in producing large heating at 4 to 5 km altitude, just above the low-level clouds which could even locally affect the environmental lapse rate (Babu et al., 2011). Combining data from ground-based and airborne measurements over oceanic regions adjoining India during the ICARB, Moorthy et al. (2009) have demonstrated the large differences in the atmospheric heating rates due to aerosols in the lower free troposphere, when height resolved scattering and absorption coefficients are used. However, their vertical measurements were confined to about 3 km above ground, over the

peninsular landmass and nearby oceans and also limited to pre-monsoon seasons only. In these studies, in-situ measurements were limited to about 3 km (Babu et al., 2010; Babu et al., 2016), primarily due to technical limitations of the aircraft sampling system. During 2009, airborne atmospheric measurements of aerosol vertical profiles extending up to 6 km asl were carried out over different parts of India as part of Cloud Aerosol Interactions and Precipitation Enhancement EXperiment (CAIPEEX), which revealed the presence of elevated aerosol layers during both pre-monsoon and monsoon (Padmakumari et al., 2013a).

During the monsoon, ground-based (or even satellite borne) measurements of AOD and extinction coefficient profiles are limited due to extensive cloudiness and precipitation. Most of the existing measurements were made during the winter/ pre-monsoon period (January to April) and hence limited measurements on the vertical distribution of aerosol characteristics just before the onset of the monsoon and during its evolving phase, especially over the central Indian regions, where the aerosol abundances and type undergo rapid changes due to changes meteorological conditions. Though the CAIPEEX campaign

addressed to this to some extent (Kulkarni et al., 2012) only its phase -I (2009) covered the evolving phase of the monsoon. During the phase-I elevated aerosol layers were observed (mainly in the 2-4 km region) over the Gangetic Plains, which led to large increases in the aerosol and CCN concentrations (Padmakumari et al., 2013b). However, the main focus of CAIPEEX was to understand the aerosol-cloud-precipitation linkages from the perspective of precipitation enhancement and as such aerosol properties relevant only towards this have been explored.



To reduce the uncertainties in the aerosol-cloud monsoon interactions it is required to bring together different datasets of aerosol properties from India, including in-situ measurements, ground-based and space based remote sensing data, and aircraft measurements. In addition, it is important to measure key aerosol physical and chemical properties required to assess the regional and vertical distribution of aerosol-induced atmospheric heating, by conducting campaign mode measurements

involving in situ measurements of aerosol properties using aircraft, supplemented with ground based and space based measurements across the Indian domain, with special focus on the Indo-Gangetic Plain and northern Bay of Bengal, immediately before monsoon onset and during its development.

The joint Indo-UK field experiment, SWAAMI was formulated to meet the above goals. During this campaign, extensive airborne measurements of aerosols were made in the altitude range from very close to the surface to as high as 8 km, using

UK's BAe-146-301 atmospheric research aircraft (Highwood et al., 2012; Johnson et al., 2012) over the Indian domain, during June-July 2016 covering the pre-onset and evolving phases of the ISM (Brooks et al., 2018),  perhaps for the first time in this detail.  This was closely preceded by airborne measurements of aerosol properties across the IGP aboard an Indian aircraft up to an altitude of ~3.5 km (Vaishya et al., 2017). In this paper, we report the vertical distribution of extinction coefficients and single scattering albedo during SWAAMI campaign and discuss the climate implications.

**2 Airborne Measurements and Data**

The airborne measurements were carried out aboard the BAe-146-301 atmospheric research aircraft operated by the Facility for Airborne Atmospheric Measurements (FAAM). The strategy was to make in situ measurements of aerosols and meteorological fields, for the first time from near the surface to ~8 km altitude, just prior to the onset of Indian summer monsoon over the Central Indian plains and during its progress to the active phase; over the landmass and adjoining oceans.

While the observations in the Indo-Gangetic Plain (IGP) and Western India, which covered a range of surface types in the South Asian region, focused on the transition of aerosol properties from dry to wet climatic zones, the observations in the peninsular Indian region focused on the transitions from land to ocean and across the orography (of the Western Ghats). The airborne measurements were carried out from five locations (base stations); Lucknow (LCK - 26.84 °N, 80.94 °E, 126 m asl) in the central IGP, Jaipur (JPR - 26.91 °N, 75.78 °E, 431 m asl) in the western IGP; Nagpur (NGP - 21.14 °N, 79.08 °E, 310

m asl) Central India; and over the Arabian Sea (AS - 13 °N, 72 °E) and Bay of Bengal (BoB - 11 °N, 84 °E,) from the base station Bengaluru (77.59 °N, 12.97 °E, 920 m asl) in the southern Peninsula (Figure 1). Of these, the measurements over LCK covered periods just prior to the onset of monsoon, as well its active phase, while the measurements in the southern peninsula and oceanic regions were carried out during the evolving phase of the monsoon. The details of the measurements from each base station, the regions covered during the flight and the period of measurements with respect to the phase of the monsoon

are summarized in Table 1. More details of the flights during the campaign are provided in the supplementary material (Table S1 and Figure S1). The center of operations for phases 1 and 3 was Lucknow and that for phase-2 was Bengaluru. The first phase measurements covered regions around Lucknow, as well as east-west transects across the IGP covering Jaipur (west)



and Bhubaneswar east) before moving over to Bengaluru, via Nagpur. The second phase focused on monsoon and the measurements were made over the Arabian Sea (western part of Indian peninsula) and Bay of Bengal (on the eastern part). The third phase covered regions to the west of Lucknow (near Jaipur) and southeast of Lucknow (near Bhubaneswar) similar to the first phase and in addition measurements were made near Ahmedabad and over Ganga en route Lucknow to Delhi (as far as Kachhla).

During the measurements, the aircraft moved at a typical ascend rate of 5.5 m s$^{-1}$ and descended at a rate of 6.5 m s$^{-1}$ making measurements at a vertical resolution of ~7 m. Thus, during its ascent from near surface to ~7 km during a profiling, the aircraft covers a horizontal distance of ~130 km with a horizontal velocity of ~100 m s$^{-1}$. In all, the aircraft made 22 dedicated scientific flights spanning approximately 100 hours in three phases: Phase-1: 11-Jun-2016 to 13-Jun-2016 (3 flights); Phase-2: 21-Jun-2016 to 30-Jun-2016 (10 flights); and Phase-3: 02-Jul-2016 to 11-Jul-2016 (9 flights). A list of the aerosol instruments aboard, the parameters retrieved from the measurements, the relevant reference to the principle of instrument and data deduction details, general aircraft data and met data is provided in Table 2. The locations of vertical profiling are marked in Figure 1, along with the important base stations on the ground and the phases of the campaign. The detailed flight track for the three phases are given in the supplementary material. The symbols indicate the phase of measurement and the square boxes over symbols indicate the availability of data from surface to > 4 km.

Measurements using the Particle Soot Absorption Photometer (PSAP) from Radiance Research are used to estimate aerosol absorption coefficients at 567 nm. The measurements were made under ambient pressure and the PSAP data was corrected for pressure, spot size, flow rate and the presence of scattering particles on the filter following Bond et al. (1999) and Ogren (2010). The scattering coefficient required for this correction was taken from the nephelometer. The nephelometer (TSI-3563) gives the scattering coefficients at three wavelengths (450, 550 and 700 nm). The scattering coefficients were corrected for the angular truncation (due to the measurement angle being limited to the range 7° to 170°), non-Lambertian nature of the light source and the dependence of the scattering on the particle size following Anderson and Ogren (1998). In addition to these, vertical distribution of static pressure from the aircraft Reduced Vertical Separation Minima (RVSM) system, temperature from Rosemount de-iced temperature sensor and water vapour from the Water Vapour Sensing System Version 2 (WVSS-II) fed through a Rosemount inlet are also used in this study. The WVSS-II, which uses a 1.37 μm laser, is used to estimate the atmospheric water vapour. This wavelength is chosen as it is not absorbed by ice crystals or aerosols but it can be scattered at cold dry regions of the upper troposphere. To ensure reliable performance, the inlet is aerodynamically designed to prevent particles from reaching the sensing cavity (Fleming and May, 2004; Vance et al., 2015). More details are available in the FAAM website https://www.faam.ac.uk.

The absorption coefficient at 567 nm, estimated using the PSAP, is extrapolated to 550 nm (where the Nephelometer measurements are available) using the inverse dependence (Angstrom exponent; (Ogren, 2010)). The corrected absorption coefficient ($\sigma_{ab}$) at 550 nm is given by



$$\sigma_{ab}^{550} = 0.873 C_{spot} \, C_{flow} \, 0.98 \frac{\sigma_{ab}^{567}}{k_2} - \sigma_{sca}^{550} \frac{k_1}{k_2} \tag{1}$$

where 0.873 accounts for the difference in spot area of the PSAP filter spot and the manufacturer's reference instrument, $C_{spot}$ = 1.186 and $C_{flow}$ = 0.909 are the correction factors for PSAP filter spot area and flow respectively specific to the instrument used, 0.98 is the adjustment factor used to convert the $\sigma_{ab}$ measured at 567 nm to $\sigma_{ab}$ at 550 nm assuming an inverse wavelength

dependence $(567/550)^{-0.5}$ = 0.98, $\sigma_{ab}^{567}$ and $\sigma_{sca}^{550}$ are the uncorrected absorption and scattering coefficients at 567 nm and 550 nm respectively and $k_1$ = 0.02 and $k_2$ = 1.22 are coefficients of the empirical relation suggested by Bond et al. (1999) to remove the effect of aerosol scattering from the absorption coefficient measurements.

The scattering coefficients ($\sigma_{sca}$) obtained from the nephelometer are corrected for angular truncation error and errors arising due to the dependence of light scattering on particle size. This is done following Anderson and Ogren (1998) and assuming no

size cut. In practice the nephelometer does not represent the full scattering coefficient from dust due to the limitation set by the inlet. Despite the inlet having a 50% cut-off efficiency at 3 μm, (Highwood et al., 2012) the assumption of no size-cut was considered for the correction as both sub-micron and super-micron particles are simultaneously observed by the nephelometer. The SSA is calculated using the corrected scattering and absorption coefficients at 550 nm the SSA is estimated as

$$SSA = \frac{\sigma_{sca}}{\sigma_{sca} + \sigma_{ab}} \tag{2}$$

Filter based absorption measurements generally overestimate absorption by up to 45% and can lead to underestimations in the SSA (up to 0.07) (Davies et al., 2019). The radiative effects of these aerosols are estimated using a radiative transfer scheme taking into account the vertical distribution of extinction coefficient, SSA, pressure, temperature, water vapour and ozone.

**3 Synoptic Meteorology during the Campaign**

The year 2016 experienced delayed monsoon onset; with onset at the southern tip of the peninsula occurring on 8th June

(instead of the normal onset date of 1st June). Following the delayed onset, the monsoon reached Nagpur and Lucknow by the 19th and 21st June respectively and covered the entire Indian region only by 13th July (which climatologically happens towards the end of June). The advance of the southwest monsoon is shown in Figure 1 (red line). Despite this delay, the rainfall for the season was near normal; about 97% of the long-term average for the season (Purohit and Kaur, 2016). The mean wind field on selected days during the progression of the monsoon throughout the campaign period are shown in Figure 2. The winds

were strong during the start of the campaign (June-11) weakened by June-19, before picking up and reaching the peak values by June-28 and again weakened by July-7. The rainfall during the season was close to the long term average and had two long deficient spells of more than 10 days (Purohit and Kaur, 2016). But during the period of the campaign the rainfall did not show





much deviation from the normal. The maximum rainfall was received during the first week of July with nearly 50% excess rain during this period mainly in central, eastern and northeastern India.

## 4 Results and Discussions

### 4.1 Vertical profiles of SSA and $\sigma_{ext}$: Spatial variation

The mean altitude profiles of extinction coefficient ($\sigma_{ext}$), scattering Angstrom exponent ($\alpha_{sca}$) and SSA, their spatial variation across the mainland and oceans, as well as temporal changes from just prior to the onset of monsoon to peak monsoon activity are presented in the different panels of Figure 3. In the figure, panels 3a, 3c, 3e, 3g, 3i and 3k represent the vertical distributions of $\sigma_{ext}$ (bottom axis) and $\alpha_{sca}$ (top axis) while panels 3b, 3d, 3f, 3h, 3j and 3l show the vertical distribution of SSA. The y-axes in all the panels represent the height in km asl. Panels 3a-3b and 3g-3h represent the conditions just before the onset of monsoon

over Central IGP (Lucknow) and Central India (Nagpur) respectively, while the other panels show the features during monsoon over Central IGP (LCK (panels c-d)), the northwestern India (Jaipur (panels e-f)) and the oceanic regions Arabian Sea (panels i-j) and Bay of Bengal (panels k-l). Panels 3 c-d show the features at LCK during the peak of the monsoon activity and thus characterize mostly the changes brought about by the monsoon rains. The following results emerge:

- Prevalence of strongly absorbing aerosol layers over most of the locations, as revealed by the low values ($\leq 0.9$) of
SSA. Profiles of SSA show sharp decrease in the altitude ranges 1 - 2.5 km and 1.5 - 3 km respectively over Lucknow and Nagpur, especially just prior to the onset of monsoon, increasing at higher altitudes. In contrast, SSA over Jaipur (having the strongest influence of dust, of all the regions in the figure) keep decreasing steadily with altitude up to 6 km reaching values < 0.9 above 5 km. An elevated layer of enhanced absorption is observed over the Arabian Sea near 2 km, while highly absorbing aerosol layer is observed over the Bay of Bengal at around 1 km and between 3
and 4 km; with SSA going as low as 0.7. Such strongly absorbing aerosol layers at higher altitudes would have stronger implications for aerosol induced atmospheric heating.

- Extinction coefficient, $\sigma_{ext}$, decreases near-exponentially with altitude over most of the mainland, while over the oceanic regions of the Arabian Sea and Bay of Bengal, an increase in extinction is indicated above 2 km, attributed to elevated layers of aerosols. These layers appear to be stronger over the Arabian Sea than over Bay of Bengal. In
general, highest values are observed within 3 km from the surface where the aerosol abundance is more.

- The scattering Angstrom exponent ($\alpha_{sca}$) indicates a decline in the relative dominance of fine mode particles at altitudes above 2 km over the entire region ($\alpha_{sca}$ <1). Over the central Gangetic plains (LCK) strong accumulation abundance is indicated closer to the surface (below 1 km).

- During active phase of the monsoon, higher $\alpha_{sca}$ values occur in the entire profile over Lucknow, (Fig 3c) suggesting
the wet removal of coarse particles by the rain.



Looking closer to the details, we see the presence of a strongly absorbing region (layer) over Central IGP (represented by Lucknow) in the altitude region 1 to 2.5 km (marked by a dashed oval in figure 3b) just before the onset of monsoon, where the SSA drops to as low as 0.7 from its nearly steady higher values (~0.93) below. Above this region, the SSA increased steadily to reach ~0.97 above 5 km. Notwithstanding this, the extinction coefficient decreased rather monotonically from >

100 Mm$^{-1}$ near the surface to reach ~ 20 Mm$^{-1}$ near 5 km. The variation of $\alpha_{sca}$ with altitude indicates the abundance of sub-micron mode particles ($1.2 < \alpha_{sca} < 1.8$) in the strongly absorbing layer in the altitude range 1 to 2.5 km, and a decline in the fine mode dominance above and below this altitude. The 7-day back-trajectories (Figure 4a) arriving at 2 km (absorbing layer) and 6 km (weaker absorption) over LCK, show distinctively different source region influences; respectively from the Persian Gulf, across the northern Arabian sea, passing across the Indian (Thar) desert and from the northern regions of the Middle

East. The mineral dust aerosols from the Gulf regions and Indian desert as well as the alluvial soil over the IGP are known to have higher absorption (Moorthy et al., 2007) due to higher hematite content (Chinnam et al., 2006) in the soil and depict a brown colour compared to the more white African dust. These are also of finer size. This might be, at least partly, responsible for the observed lower SSA and higher $\alpha_{sca}$ in this layer. Additional possibilities are mixing of the absorbing dust (with high porosity) with locally emitted absorbing species such as BC. During the peak monsoon activity (Figure 3c) the same region

shows a rapid depletion of aerosols above 3 km. Even within the first 3 km, aerosols are now far less absorbing (Figure 3d) with SSA hovering around 0.9, until 2.5 km. The decrease above that could be at least partly due to the very rapid decrease in the extinction coefficient due the depleted aerosol concentration. Interestingly, this region is dominated by submicron aerosol, $\alpha_{sca} > 1.5$ throughout the column up to ~2.5 km, during the monsoon, showing probable washout of coarse mode aerosols by the monsoon rains. Airborne measurements during CAIPEEX has shown cloud base altitude in the range 0.65 to 1.17 km in

the IGP, with a large vertical extent (Konwar et al., 2012). There is a large shift in the back-trajectories arriving at 2 km and 6 km over LCK (Figure 4b) during this period. They are almost confined over the Indian landmass, with little advection of mineral dust from the west and hence are conducive for advection of anthropogenic aerosols (mostly produced locally) to the receptor site leading to fine mode dominance ($\alpha_{sca} > 1.5$) over this region. Examining our values with those reported earlier for this region, based on airborne measurements, Earlier observations over this region have reported a columnar (up to 3 km

altitude) mean SSA of 0.86 at 520 nm over Lucknow during pre-monsoon period (Babu et al., 2016) which is just a shade higher than our value of 0.83 ±0.08 (for the altitude range 0-3 km for the same season).

Above the Central Indian region (Nagpur, Figure 3h), the broad features of the vertical profile of SSA observed over the IGP prior to onset of monsoon recurred, however, with some difference. It is important to note that even at Nagpur, the profiling was just prior to the onset of monsoon there, which occurred only by June 19 (Figure 1). While the main features of the profile

(higher SSA closer to the ground, an elevated layer of absorbing aerosols above the boundary layer where the SSA drops to lower values followed by a rapid increase to higher altitudes) remained quite similar, the magnitudes were higher than those seen over the IGP. Near to the surface (within the boundary layer) a uniform SSA of 0.95 is seen, extending up to almost 2 km; above which there is a layer of reduced SSA (~0.89) extending up to 3 km. Above that, SSA increases rapidly, approaching values as high as 0.98 at 5 km altitude. The column integrated SSA (up to 5 km) increases from $0.88 \pm 0.08$ at LCK to $0.93 \pm$



0.02 at NGP suggesting an overall change to predominantly scattering aerosols as we move southward of the IGP. Similarly, the magnitudes of the extinction coefficients near the surface are lower than those seen over the IGP (Figure 3g), decreasing more slowly with altitude, and becoming comparable to the IGP values above around 4 km. Scattering Angstrom exponent shows values remaining consistently below 1.0. Furthermore, $\alpha_{sca}$ decreases with increase in altitude, reaching values as low as 0.5 at around 5 km, showing very small fraction of sub-micron aerosols in the composite aerosol system. The back trajectories (Figure 5) show strong marine aerosol advection in the lower altitudes (1.5 km), while aerosols advected from Persian Gulf passing over the northern Arabian Sea influence the 2-3 km region which shows enhanced absorption. Above that (at 4 km) the influence of the Thar Desert becomes significant, the influence of marine aerosols become negligible and the SSA values become comparable to that at LCK. This is the region where there is significant coarse mode dominance ($\alpha_{sca}$ ~0.5) and high values of SSA (~0.95). The trajectories reaching NGP at 5 km trace back to the northern Indian landmass and the SSA values are again comparable to those observed over LCK at this altitude.

By the start of second phase of the campaign, which mainly covered the oceanic regions on either side of the peninsula (Arabian Sea on the west and Bay of Bengal on the east), monsoon winds have fully established over the regions and the rainfall has covered the entire domain. The measurements in this phase were limited to ~4 km asl partly due to the focus on cloud characteristics during this period and partly due to the poor quality of data above this altitude due to extensive cloudiness.

The vertical structure of extinction and SSA show distinctive differences over the two oceans, primarily attributed to the differences in the advected components. Over the Arabian Sea, the aerosols near the surface are almost entirely scattering in nature (SSA very close to 1, Figure 3j). The SSA decreases towards higher altitudes, with a well-defined and strongly absorbing layer occurring between 1.5 and 2 km, where the SSA drops to as low as 0.8., followed by a gradual increase to reach ~0.95 at 4 km, indicating prevalence of absorbing aerosols aloft. This feature is quite similar to what was seen over the IGP (LCK), except that the SSA values are slightly higher over the ocean, probably due to absence of local emissions of absorbing species such as BC or fine dust and presence of sea-salt. However, most important to note is the perseverance of the rather strongly absorbing layer in the altitude range 1.5 to 3 km over the entire region (IGP, Central Peninsula and Arabian Sea, in which the particles are more coarse in nature as revealed by the lower value of the Angstrom exponent (values ~0.6 within the first 2 km range, while the values decrease to ~0.5 near 2-3 km region where the lowest values of SSA are observed). The values again increase to ~0.6 above this region. Examining the back-trajectories over this region (Figure 6a), it is interesting to note that those arriving at 2 km altitude (where the strongly absorbing layer is seen) pass across the Somali and Horn of Africa regions where the absorption efficiency of dust has been observed to be very high (Deepshikha et al., 2005). The SSA at 500 nm over New Delhi (which also lies in the IGP) where both desert dust and anthropogenic aerosols has been shown to decrease from 0.84 in March to 0.74 in June (Pandithurai et al., 2008). Similar values of SSA were also reported by Ge et al. (2010) from Zhangye, northwest China (a semi-arid area) where the mean SSA at 500 nm was reported to be 0.75 ±0.02 during April - May.

Over the BoB (Figure 3k), the most striking feature are the low values of SSA (0.9) even near the surface which are in sharp contrast with Arabian Sea having an SSA ~0.98 near the surface. There are two absorbing layers; one just below 1 km (SSA





~0.8) and another strongly absorbing layer above 3 km (SSA ~0.7) with higher values (~0.9) in between. The extinction coefficient (Figure 3j) decreased from ~45 Mm$^{-1}$ at the surface to a lowest value of ~15 Mm$^{-1}$ at 2 km above which $\sigma_{ext}$ values sharply increase to ~30 and remain more or less constant (weak decrease) above this altitude. The $\alpha_{sca}$ remains <1; indicating the dominance of coarse mode particles throughout the column. The values are close to 1 near the surface, while the lowest

values ($\alpha_{sca}$ ~0.6) are observed in the 2-3 km range, again within the layer of strong absorption as seen in the earlier cases. Again the $\alpha_{sca}$ values increase to ~1 above 3.5 km where an absorbing layer is observed (SSA ~0.7). It is interesting to note that the back trajectories (Figure 6b) reaching this region at 1 km, originate close to the Persian Gulf region. Coincidentally, low SSA was observed (near 2 km) over the Arabian Sea also, when the trajectories originated from this region. It is likely that both these regions receive the same aerosol type from the same region. This is also supported by the occurrence of this

layer at lower altitude (~1 km) over BoB (compared to ~2 km over the Arabian Sea) and with extinction coefficient of a comparable magnitude. It is interesting to notice the elevated layer of higher extinction coefficient (between 3.5 and 4 km) over BoB, has similar properties to that seen over the Arabian Sea, but is less intense. These difference in the positions of the sharp decrease in SSA and decrease in the magnitude of the extinction coefficients are likely to be resulting from washout by the extensive monsoon rains, aided by sinking motion of air while reaching BoB. The role of the low level monsoon jets across

the Arabian Sea in maintaining the elevated absorbing aerosol layers over the whole of southern peninsula has been reported recently by Ratnam et al. (2018).

Despite these linkages, on an average the columnar and near surface values of SSA over BoB remains much lower than that over the Arabian Sea. The column integrated values of SSA over BoB and AS are 0.84 ±0.07 and 0.89 ±0.04 respectively during this campaign. This is perhaps the first-time the vertical structure of SSA over oceanic regions around the Indian

peninsula has been reported. However, a few ship board measurements earlier (mostly during winter and pre monsoon) have shown that aerosols over BoB are generally more absorbing (lower SSA) than over the Arabian Sea, and also show a sharp north-south gradient (Nair et al., 2008; Satheesh et al., 2010). The SSA values reported by Nair et al. (2008) based on extensive cruise measurements during the pre-monsoon season are very close to our values near the surface. Back-trajectory analysis in Fig 6 (b) shows significant advection, across the peninsular landmass at lower levels, supporting the higher extinction

coefficients and lower SSA. The $\alpha_{sca}$ values in the range 0.5-0.8 suggest the relative dominance of fine mode particles is weak over this region and the $\sigma_{ext}$ values are found to increase when airmasses arrives from continental regions. The additional continental input as the airmass transits across the peninsula is likely to lead to the lower SSA over BoB, consistent with several previous ship-board measurements (Nair et al., 2008; Babu et al., 2012).

During the final phase of the project when the Indian monsoon was in its active phase in July, the measurements were repeated

over the IGP at locations near Jaipur (JPR) and Lucknow (LCK). Jaipur is located in northwest India, characterized by arid/semi-arid terrain and is weakly influenced by the monsoon, while LCK now well within the monsoon trough region. The vertical variation of SSA over JPR still shows presence of absorbing aerosols (mostly dust), with SSA decreasing gradually from its value close to 0.95 near the surface to ~0.88 at 6 km altitude, with a weak dip around 3.5 km where SSA goes down to ~0.85 (the shape resembles the IGP profile prior to the onset of the monsoon; but the magnitudes are higher). The back





trajectories (Figure 7) show two distinct branches with one arriving from the northwest from continental region or from northern Arabian Sea and another from Southeast Asia traveling across the IGP region. The low values of $\alpha_{sca}$ (<1) show again the low concentrations of fine mode aerosols over this region at all altitudes up to 7 km. At Lucknow, as stated earlier, significant washout leads to very low extinction above 3 km, while at the lower altitudes, the SSA has increased compared to

the pre-monsoon with values remaining around 0.9 up to 2 km and decreasing above to be less than 0.8 at 3 km.

In summary, all the measurements have revealed a vertical heterogeneity in SSA over the IGP, AS and BoB, with indication of lower SSA (higher aerosol absorption) in the lower free troposphere (2 to 3 km); which while would remain below the convective clouds during the pre-monsoon, but just above the low-level clouds (cloud base around 1 km during monsoon season, (Konwar et al., 2012)) during the peak monsoon activity. Back trajectory as well as spectral scattering suggest these

layers to be dominated (if not entirely due to) by advected absorbing dust. Such absorbing layers at higher altitudes would produce more warming of the atmosphere (due to reduced air density) and would have implications for clouds and precipitation. The radiative implications are examined below.

## 4.2 Radiative forcing estimates using measured extinction and SSA profiles

The radiative effects were estimated using the measured vertical distribution of aerosol properties ($\sigma_{ext}$, SSA) and atmospheric

parameters (pressure, temperature and water vapour) from 6 locations LCK, NGP, AS, BoB, BBR and JPR. The instantaneous atmospheric forcing corresponding to the local noon were calculated. In most of the existing studies the aerosol forcing is estimated assuming a constant value of SSA throughout the atmosphere and very often SSA values obtained from either near surface measurements (Ganguly et al., 2005; Moorthy et al., 2009) or columnar SSA retrieved for sky radiance measurements (Dubovik et al., 2000; Choi and Chung, 2014) are used. However, these would not consider the vertical heterogeneity, which

is important. To examine this aspect, we estimated the atmospheric radiative forcing due to aerosols (difference between the surface and TOA forcing) using the measured SSA and extinction profiles along with the concurrent atmospheric parameters (measured during the experiment) using the SBDART radiative transfer scheme (Ricchiazzi et al., 1998). We also evaluated the atmospheric forcing by replacing the SSA profile with a single SSA value, computed as the weighted average of the profile.

$$SSA = \frac{\sum_i SSA_i.height_i}{\sum_i height_i} \qquad (3)$$

The results are shown in Figure 8, in which the black bars represent the values estimated using the weighted average SSA and the red bars represent the values obtained using the measured SSA profiles. It can be clearly seen that, with exception of the Arabian Sea, the use of a single SSA value (even after giving the weightings for the vertical variation) results in forcing estimates that are lower by 2-6 Wm$^{-2}$ than the estimate making use of the actual SSA profile; with higher differences over the IGP. As the monsoon progresses to its active phase, the atmospheric forcing decreases significantly by as much as 10 Wm$^{-2}$





from the values that persisted just prior to the onset of monsoon. Nevertheless, the forcing is significant over the IGP with the monsoon values at LCK being nearly same as the pre-monsoon value over NGP. However, most interestingly, the pattern is significantly reversed over the Arabian Sea, with higher forcing value when a constant SSA is used. It has to be recalled that this location has absorbing aerosol layers at high altitudes and also high values of extinction coefficients (higher than that near

the surface) at higher altitudes.

### 4.3 Estimation of Heating rates and its vertical profiles

To examine the possible effects of the elevated aerosol layer and elevated heating on atmospheric heating rates, we have estimated the vertical distribution of the aerosol induced atmospheric heating rate for all the locations, using height resolved SSA and a constant weighted mean SSA. The aerosol induced heating rates $(\frac{\partial T}{\partial t}(K\ day^{-1}))$ were calculated as

$$\frac{\partial T}{\partial t} = \frac{g}{C_p} \frac{\Delta F}{\Delta P} \qquad (4)$$

where g is the acceleration due to gravity, $C_p$ is the specific heat capacity of air at constant pressure and $\Delta F$ and $\Delta P$ are the aerosol induced layer forcing and change in pressure between the top and bottom of the layer respectively.

The estimated heating rate profiles for three locations Lucknow, Nagpur and Jaipur are shown in Figure 9. The black lines represent the cases where mean SSA is used and the blue lines represent the cases where the SSA profile is used for the heating

rate estimation. Figure 9 a & b represent the heating rate profiles prior to the onset of monsoon, while Figure 9 c & d represent the heating rates profiles during the active phase of the monsoon. The striking feature revealed is that the elevated heating layers which exist during both pre-monsoon and monsoon periods when the height resolved measured SSA has been used, are totally absent when a single weighted mean SSA was used (despite both being location specific and season specific). This implies that wherever the weighted mean SSA is greater than the height-resolved (actual) SSA there is an underestimation in

the layer heating and vice versa. Hence, the heating induced by elevated absorbing layers would not be captured properly by simply using extinction coefficient profiles and a single SSA value, even if it is station-specific and season specific. Hence, it emerges that the higher forcing over the Arabian Sea when a single SSA value is used (in contrast with other locations) is due to the high extinction coefficients at higher altitude (Figure 3i) and the low SSA at high altitude (Figure 3j). While using the profiles of $\sigma_{ext}$ and SSA, the region where the SSA is less does not have significant increase in heating due to low $\sigma_{ext}$ while

the region where $\sigma_{ext}$ is high has low heating due to high SSA. This very clearly emphasizes the importance of using height-resolved extinction coefficients and SSA for improving the atmospheric heating due to aerosols and assessing their climate implications.

Similarly from Figure 9a it can be seen that the large heating rate below 1 km, which resulted because of using the weighted mean SSA in the region where high extinction coefficients were present (Figure 3a), instead of using realistic SSAs; which





when used produce much higher heating rates (blue line). Using the realistic SSA results in the peak in the heating rate occurring close to 2 km where the highly absorbing elevated layer was present. A similar peak coincident with the absorbing layer above 2 km (Figure 3h) is seen in NGP (Figure 9b) also. At JPR (Figure 9c) and LCK (Figure 9d) also we notice an enhancement in the heating rate close to the absorbing layers. The observed height of peak heating rates match well with those

by Satheesh et al. (2009) for the pre-monsoon season. Using modelling studies Kuhlmann and Quaas (2010) also obtained a peak shortwave heating rate in the 2-4 km range in the IGP region; however, their estimates were much smaller in magnitude than those seen in this study. At LCK it can be seen that during pre-monsoon more heating occurs at higher altitudes, while during the monsoon, higher values occur near the surface, which are due to the wash out of aerosols at higher levels and persistence of near surface absorbing aerosols arising from local emissions. Even so, during peak monsoon, a weak peak of

elevated aerosol heating persists around 3 km (Figure 9d), which appears to be due to the advected dust, that manifested as a stronger peak above 3 km at Jaipur, during the monsoon (Figure 7). Such elevated heating layers may well affect the water cycle dynamics and the surface energy feedback leading to strong impact on the Indian Summer Monsoon and would support to the elevated heat pump hypothesis (Lau et al., 2006). The results from the Indian aircraft measurements Vaishya et al. (2018), which almost culminated during the Phase-1 of the FAAM, fall close to our observations for the western and central

IGP; however their measurements were limited to ~3.5 km due to limitations with the aircraft being flown in unpressurized mode. The existence of a meridional gradient in SSA and atmospheric heating, increasing gradually from the northern Indian Ocean to central India and over both AS and BoB has been reported by Satheesh et al. (2010). Using extensive shipboard measurements, Nair et al. (2013) have also reported strong meridional and zonal gradients in aerosol induced heating rates over AS and BoB across the peninsular landmass (from <0.1 K day$^{-1}$ over the south-western Arabian Sea, increasing to 0.5 K

day$^{-1}$ over the north-eastern Bay of Bengal) . However, these studies used a single SSA value for the entire altitude range, unlike the present case, which provides for the first time the vertical profiles of SSA and aerosol heating rate, its spatial variation across the Indian landmass and temporal changes during the evolution of monsoon. This information would be important in better modelling the aerosol-cloud-monsoon interactions.

**5 Summary**

Vertical distributions of sub-micron aerosol properties at five locations Lucknow (LCK), Nagpur (NGP), Arabian Sea (AS), Bay of Bengal (BoB) and Jaipur (JPR) were studied during the period June-July of 2016 using in-situ measurements made from an aircraft. This study depicts the spatial variations of aerosol properties over the Indian region immediately prior and during the onset of the monsoon. This is the first study where high resolution vertical profiles of SSA and extinction coefficients extending from the surface to ~ 6 km have been used to estimate the aerosol radiative forcing over the Indian region. The major

findings are given below.

- Strongly absorbing aerosol layers prevailed over the Indian region throughout the campaign period. These layers were mostly observed in the 1-3 km region, over the Indian mainland and the surrounding oceans. On an average, a drop



of ~ 0.2 was observed in the sub-micron aerosol SSA values near these absorbing layers, which were associated with long range transport of aerosols from the Persian Gulf, Middle East and Thar Desert.

- During the monsoon period, large spatial variation was observed in the SSA over the Indian region. In the central IGP (LCK), the near surface SSA was lower compared to the western IGP (JPR), revealing an east-to-west gradient in the SSA over this region.

- The relative dominance of fine mode aerosols declined ($\alpha_{sca} < 1$) above 2 km over the entire Indian region indicating significant influence of dust. Following the onset of monsoon, dust transport over LCK declined ($\alpha_{sca}$ increased), while dust aerosols continued to dominate over JPR. It was also observed that the coarse mode aerosols over JPR extended up to 6 km even during monsoon asserting the rather minor influence of the monsoon at removing aerosol in this region. As a result, a steady decrease in the SSA values with increasing altitude was observed over JPR during monsoon.

- Meanwhile, over the oceanic region, near surface SSA values over the BoB were lower compared to those over the AS conforming to earlier studies. We found that the low values of SSA were not restricted to lower altitudes, but extended throughout the column.

- The observed absorbing layers have significant influence on the regional radiative balance. The use of SSA profiles to estimate the aerosol forcing revealed significant difference (2-6 W m$^{-2}$) in the forcing estimates compared to those obtained by using constant SSA.

- The atmospheric heating rate profiles revealed an underestimation of the heating rates in those layers where the measured SSA was lower than the average value and vice versa. The magnitude of the difference in the estimated heating rates in each layer also depends on the extinction coefficients of these layers. This difference becomes significantly large if low SSA and large extinction coefficients coexist in these atmospheric layers. While the maximum heating rates in the absorbing layers (near 3 km) can be as high as 1 K day$^{-1}$, the underestimation in these layers can be greater than 0.5 K day$^{-1}$. Such large elevated warming layers have significant climate implications as they can perturb the onset of monsoon through their impact on the atmospheric stability. Incorporating such realistic aerosol profiles into climate models will significantly reduce the uncertainties in the modelled climatic impacts.

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



**Figure 1: The locations corresponding to the vertical profiles are marked in the figure with the symbol indicating the phase of measurement. The locations where high altitude data is available are indicated by the big square. The red lines show the position of the monsoon trough on the dates of the year 2016 shown at the ends of the lines, taken from the India Meteorological Department's website: http://www.imd.gov.in/pages/monsoon_main.php and is used to infer on the phase of the monsoon (prior to onset, active etc) for the different regions.**





**Figure 2: The mean wind field at 850 hPa during different days of the campaign period (Dee et al., 2011) with the colours representing the magnitude of the wind in ms⁻¹. The colour-bars represent the magnitude of wind speed. The arrows indicate the direction of the mean wind. The northern most position of the monsoon trough for the day is shown by the dashed red line on each panel (data**
5   **source: http://www.imd.gov.in/pages/monsoon_main.php).**



**Figure 3: The vertical distribution of extinction coefficients ($\sigma_{ext}$) and single scattering albedo (SSA) and the vertical distribution of scattering Angstrom exponent ($\alpha_{sca}$) during the SWAAMI campaign are shown for five locations, Lucknow, Jaipur, Nagpur, Arabian Sea and Bay of Bengal. Data for both pre-monsoon (a and b) and monsoon (c and d) is available only for Lucknow. The data available from Bhubaneswar was limited due to instrument failure during pre-monsoon (phase I) measurements and poor data quality during monsoon (phase III) and hence not included in the discussion.**





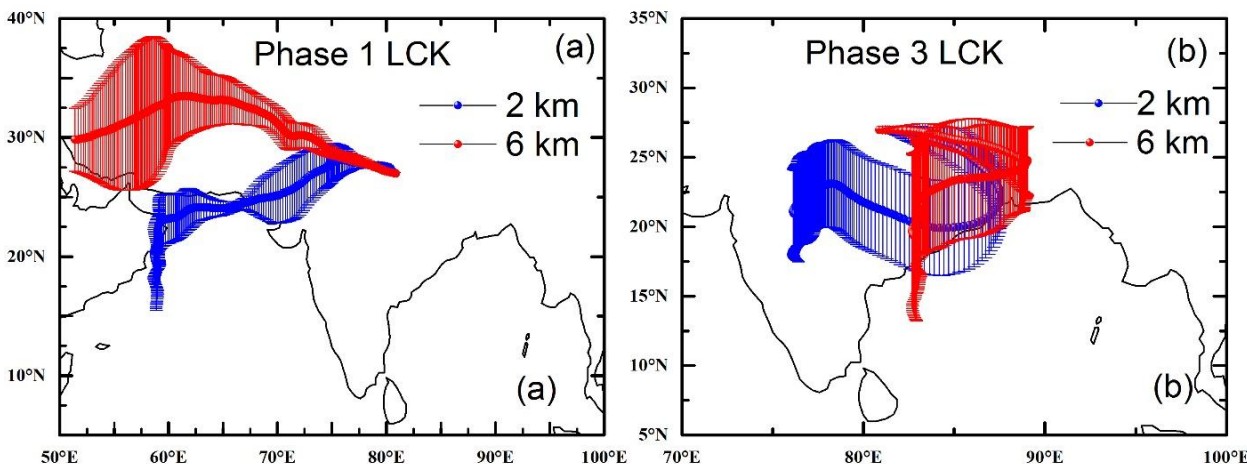

**Figure 4: Back trajectories showing the distinct air masses reaching Lucknow during, (a) pre-monsoon and (b) monsoon. The altitude indicates the height above sea level at which the airmass reaches the receptor site.**

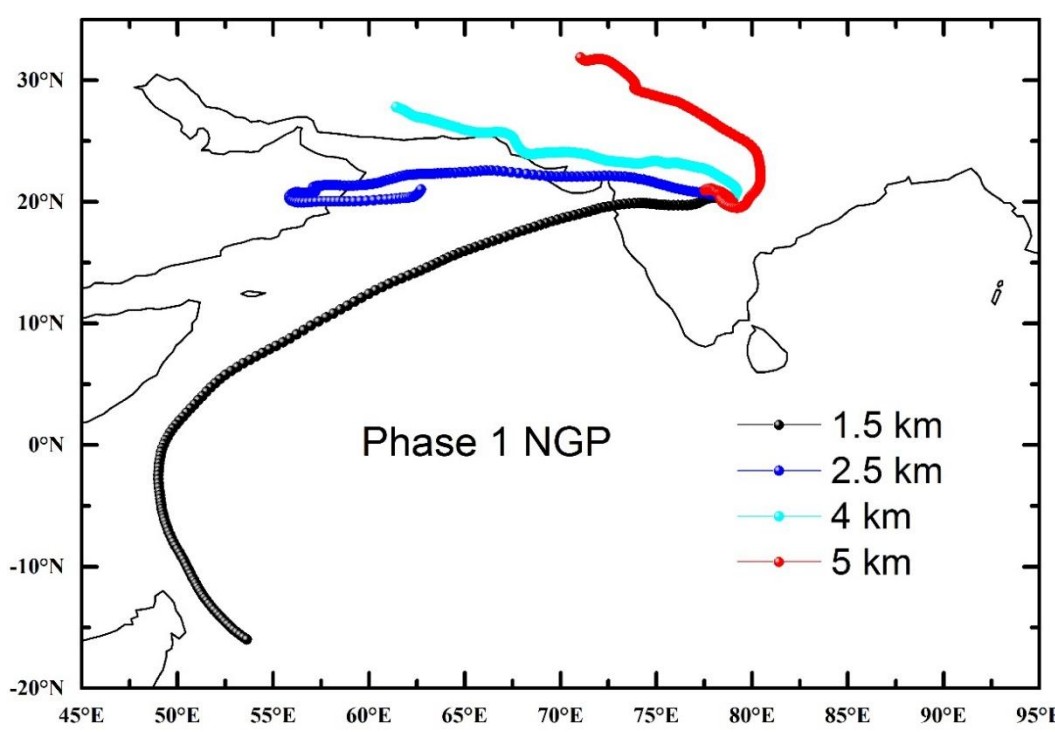

**Figure 5: Back trajectories showing the distinct air masses reaching Nagpur during the period of observation. The altitude indicates the height above sea level at which the airmass reaches the receptor site.**





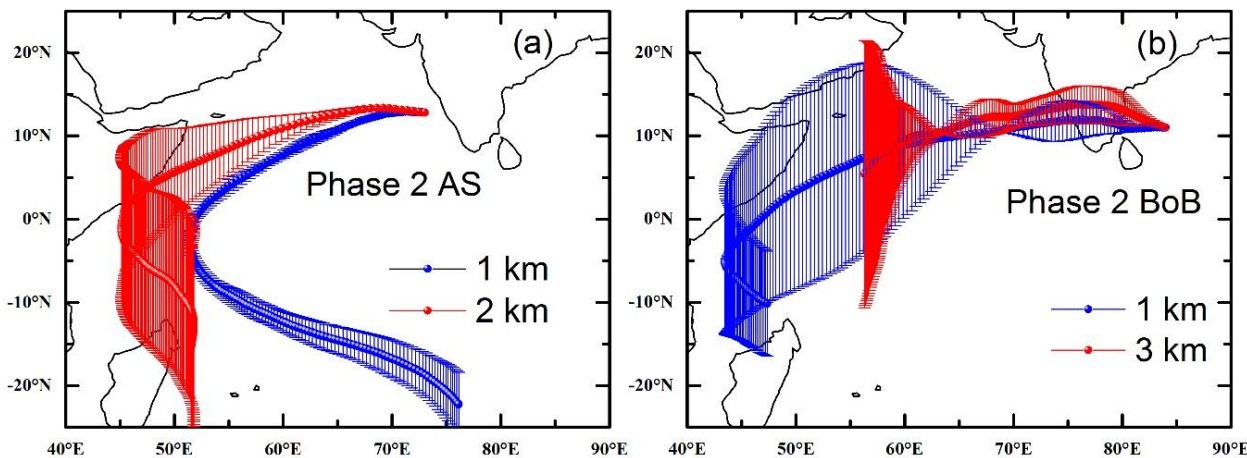

**Figure 6 Back trajectories showing the distinct air masses reaching (a) Arabian Sea and (b) Bay of Bengal during the period of observation. The altitude indicates the height above sea level at which the airmass reaches the receptor site.**

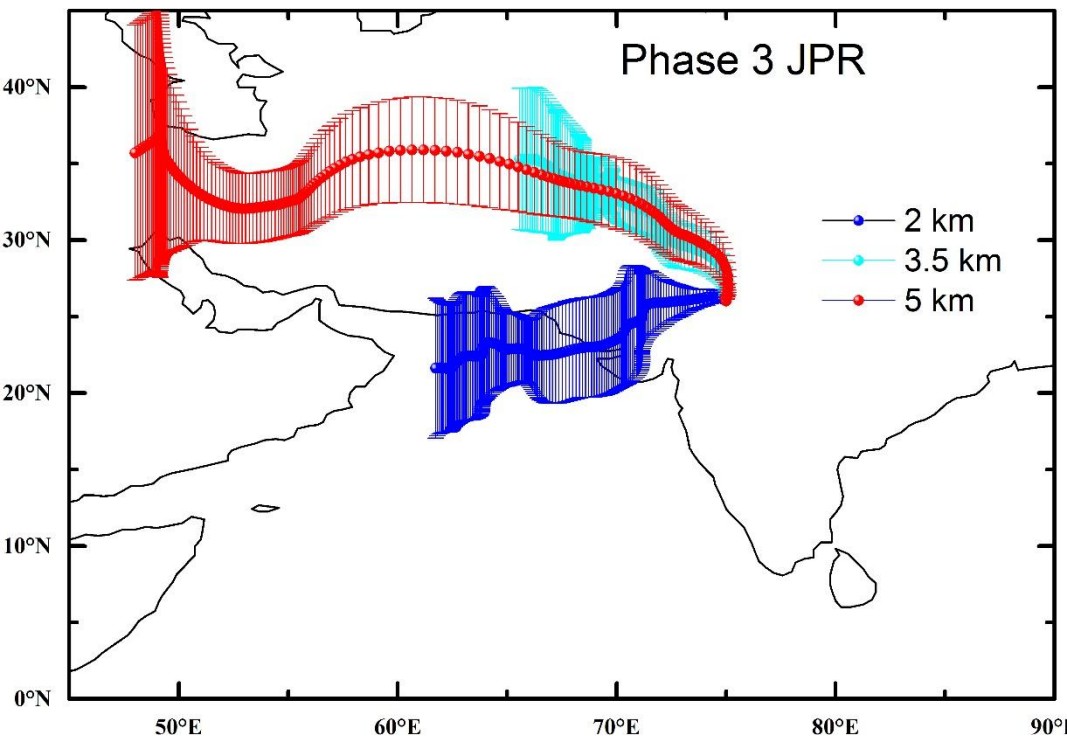

**Figure 7 Back trajectories showing the distinct air masses reaching Jaipur during the period of observation. The altitude indicates the height above sea level at which the airmass reaches the receptor site.**

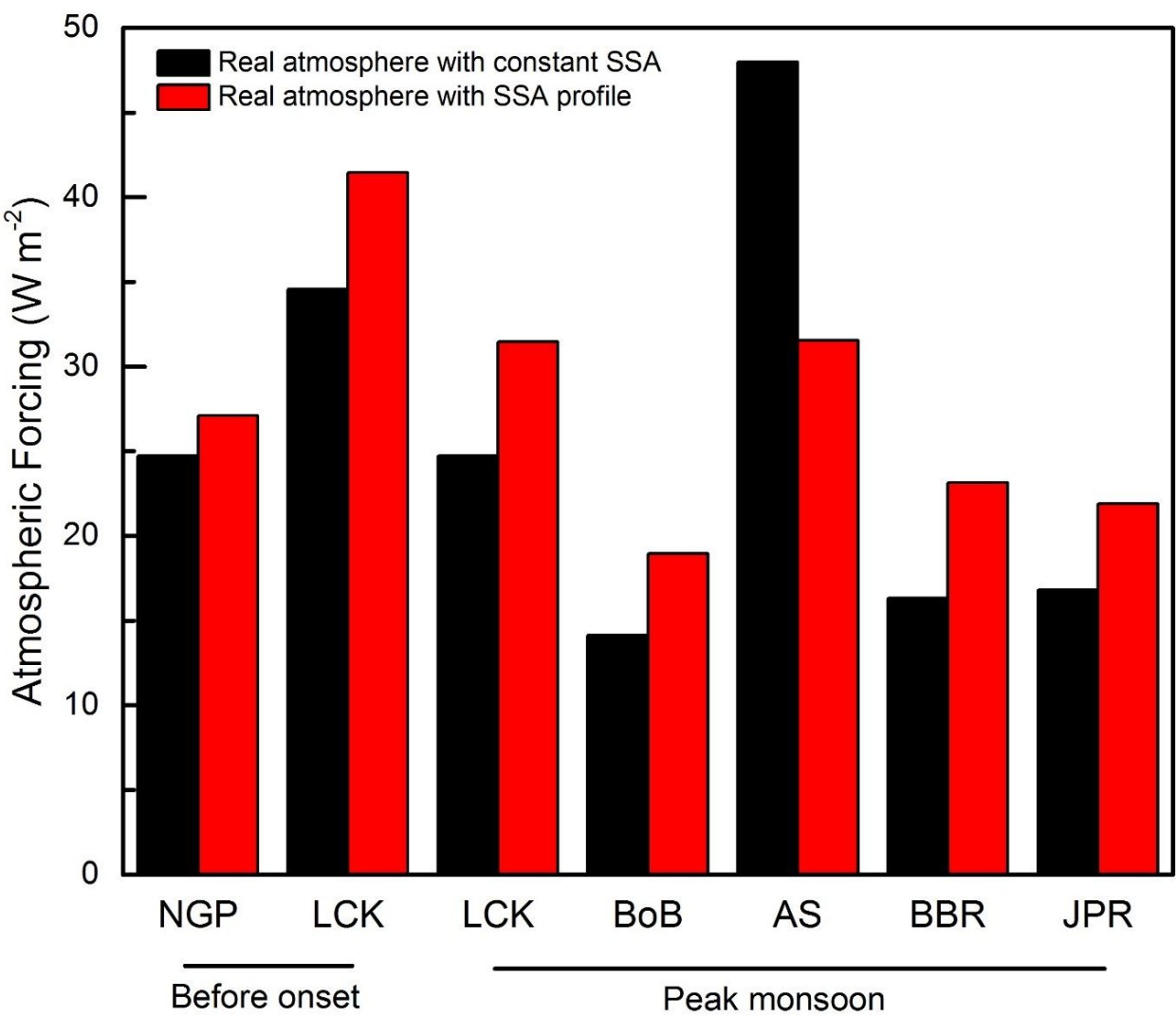

**Figure 8. The instantaneous aerosol forcing for selected profiles estimated for the local noon and averaged for the locations is shown. The first two pair of bars correspond to pre-monsoon measurements and the rest correspond to monsoon conditions.**



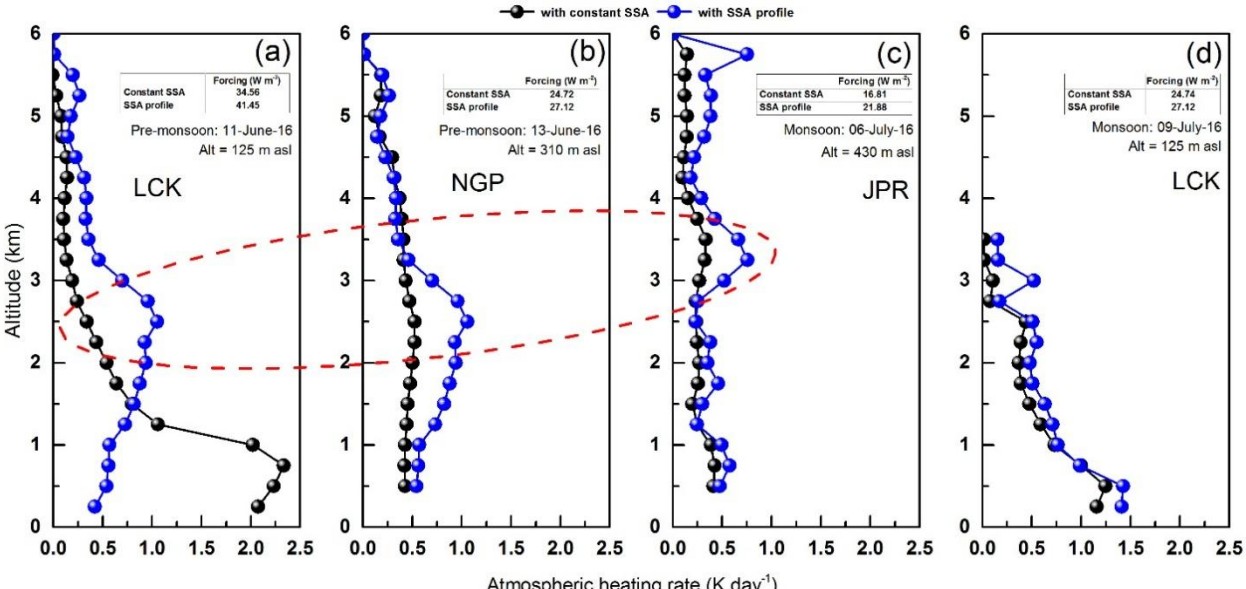

**Figure 9 Equivalent heating profiles (corresponding to local noon) for Central IGP (Left Panel) and Central Peninsula (Nagpur, second left) just before the onset of monsoon. The right panels show the profiles during active monsoon phase, on the western IGP (Jaipur) and Central IGP (Lucknow). The black lines joining the black spheres represent the heating rate profile estimated using altitude resolved extinction coefficients and columnar SSA, for the realistic atmosphere (measured pressure, temperature and water vapour) in the RT computation while the blue lines joining the blue spheres represent the heating rate profile altitude-resolved SSA values are used in place of columnar. The occurrence of the elevated layers of enhanced aerosol heating of the atmosphere is clearly seen in the latter case, emphasizing the importance of using realistic SSA profile. It is also interesting to note that the enhanced heating occurs at cloud layer altitude.**



| Base Station | Date | Regions Covered | Phase of Monsoon |
|:---:|:---|:---:|:---:|
| **LCK** | 11 - 13  June | Lucknow, Bhubaneswar, Jaipur and Nagpur | Pre-monsoon |
| **BLR** | 13 - 28 June | Bengaluru, Arabian Sea, Bay of Bengal and Nagpur | Peak monsoon |
| **LCK** | 30 June - 11 July | Lucknow, Bhubaneswar, Ahmedabad and Jaipur | Peak monsoon |

**Table 1: The period of measurement form each base station and details of regions covered during different phase of the monsoon.**



| Aerosol Instruments | Parameter | Details | Reference |
|---|---|---|---|
| PSAP | Aerosol absorption coefficient | Radiance Research<br>Wavelength = 567 nm | (Bond et al., 1999) |
| LIDAR | Elastic backscatter and depolarization | Leosphere ALS450<br>Wavelength = 355 nm | (Marenco et al., 2011) |
| Nephelometer | Aerosol scattering coefficient | TSI 3563<br>Wavelength = 450, 550, 700 nm | (Anderson and Ogren, 1998) |
| Broadband radiometer | Radiation | Eppley radiometers<br>Clear dome: wavelength = 0.3-3 µm,<br>Red dome: wavelength = 0.7-3 µm | (Haywood et al., 2008) |
| Solar Hemispheric Integrating Measurement System (SHIMS) | Radiation | Spectral radiances at 303.4 nm to 1706.5 nm | (Haywood et al., 2008) |
| Airborne Infra-Red Interferometer Evaluation System(ARIES) | Radiation | Radiance at 3.3–18 µm | (Johnson et al., 2012) |
| Passive cavity aerosol spectrometer probe (PCASP) | Aerosol optical diameter and concentration | Droplet Measurement Technologies (PCASP-100x)<br>Size range: 0.1–3 µm (depends on refractive index of aerosol) | (Highwood et al., 2012) |
| Cloud condensation nuclei (CCN) counter | CCN | Droplet Measurement Technologies (CCN-200)<br>Size range: 0.5 to 10 µm | (Trembath, 2013) |
| Aerodyne time-of-flight aerosol mass spectrometer (ToF-AMS) | Aerosol composition and mass | Aerodyne Research (AMS)<br>Size range: 50-800 nm | (Highwood et al., 2012) |
| Single Particle Soot Photometer (SP2) | Refractory black carbon aerosol mass | Droplet Measurement Technologies (SP2)<br>Size range: 70-600 nm | (Highwood et al., 2012) |



| Aircraft Instruments | Parameter |
|---|---|
| GPS aided Inertial Navigation | Latitude, Longitude, Pitch angle, Roll angle, Heading, Solar zenith, Solar azimuth, Aircraft velocity down, east and north from POS AV 510 |
| Turbulence Probe | Angle of attack, Angle of sideslip, True airspeed |
| WVSS-II | Water Vapour Measurement from Flush inlet and Rosemount inlet |
| Pitot Static System | Static Pressure, Airspeed, Altitude |
| Temperature Sensor | True air temperature from the Rosemount deiced and non-deiced temperature sensor |
| Buck CR2 Cryogenic Hygrometer | Water vapour volume mixing ratio |
| General Eastern Instrument | Dew point |

**Table 2: Details of aerosol instruments and aircraft instruments measuring various aerosol, meteorological and aircraft parameters.**

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
