# Peer review of "Vertical profiles of sub-micron aerosol single scattering albedo over Indian region immediately before monsoon onset and during its development: Research from the SWAAMI field campaign"

_Atmospheric Chemistry and Physics, 2019_

## Referee Comment (RC1) · Anonymous Referee #1 · 24 Sep 2019

The authors present single scattering albedo (SSA) profiles obtained over the Indian region during the SWAAMI field campaign in June and July of 2016. Two profiles are presented from during the Indian monsoon in addition to several profiles sampled before the onset of the monsoon. The SSA measurements are derived from measurements for absorption coefficient made by the Particle Soot Absorption Photometer (PSAP) and scattering coefficient made by TSI-3563 nephelometer. Significant variation in the SSA was observed during the campaign, with values ranging from near unity

to as low as 0.7 in one region. All profiles are explored through back trajectory, radiative forcing (RF) and heating rate calculations. The temporal, horizontal and vertical variations in SSA was attributed primarily to the prevailing wind patterns, with the most absorbing particles being advected from the Middle East, a finding consistent with previous studies (e.g. Moorthy et al., 2007). It was found that the vertical profile of SSA can have a non-negligible impact of top-of-atmosphere (TOA) radiative forcing values and that higher optical thickness and stronger absorption both correlated with larger heating rates.

Aerosol-monsoon interactions over South Asia is a very active research topic for which many open questions remain. Some key details are omitted from the manuscript but, overall, the present work appears to be a scientifically sound exploration of this topic. However, only six different profiles are presented, and only one pair of these consists of measurements made before and during the monsoon at the same location. Many other studies have explored monsoon-aerosol interactions over India using aircraft data (see Li et al., 2016 )—including an analysis of measurements from the same period as the present manuscript (Vaishya et al., 2018)—and it is not clear to me what makes the data presented in the present work particularly unique. Moreover, the supporting RF and heating rate calculations that are provided do not seem to generate novel insights. The fact that increased quantities of absorbing aerosol leads to higher heating rates is well established (e.g. Ramanathan et al., 2007). The potentially meaningful dependence of radiative forcing on the vertical profile of SSA, elucidated by the simulations, is less self-evident but has been discussed extensively elsewhere (e.g. Haywood et al., 1998; Meloni et al., 2005; Guan et al., 2010). In my opinion, the manuscript may be technically sound but the overall conclusions are not substantial enough to merit publication in high impact journal like ACP.

Moorthy, K.K., Babu, S.S., Satheesh, S.K., Srinivasan, J. and Dutt, C.B.S., 2007. Dust absorption over the "Great Indian Desert" inferred using groundbased and satellite remote sensing. Journal of Geophysical Research: Atmospheres, 112(D9).

Li, Z., Lau, W.M., Ramanathan, V., Wu, G., Ding, Y., Manoj, M.G., Liu, J., Qian, Y., Li, J., Zhou, T. and Fan, J., 2016. Aerosol and monsoon climate interactions over Asia. Reviews of Geophysics, 54(4), pp.866-929.

Kulkarni, J.R., Maheskumar, R.S., Morwal, S.B., Padmakumari, B., Konwar, M., Deshpande, C.G., Joshi, R.R., Bhalwankar, R.V., Pandithurai, G., Safai, P.D. and Narkhedkar, S.G., 2012. The cloud aerosol interactions and precipitation enhancement experiment (CAIPEEX): overview and preliminary results. Curr. Sci, 102(3), pp.413-425.

Vaishya, A., Babu, S.N.S., Jayachandran, V., Gogoi, M.M., Lakshmi, N.B., Moorthy, K.K. and Satheesh, S.K., 2018. Large contrast in the vertical distribution of aerosol optical properties and radiative effects across the Indo-Gangetic Plain during the SWAAMI–RAWEX campaign. Atmospheric Chemistry and Physics, 18(23), pp.17669-17685.

Ramanathan, V., Ramana, M.V., Roberts, G., Kim, D., Corrigan, C., Chung, C. and Winker, D., 2007. Warming trends in Asia amplified by brown cloud solar absorption. Nature, 448(7153), p.575.

Meloni, D., Di Sarra, A., Di Iorio, T. and Fiocco, G., 2005. Influence of the vertical profile of Saharan dust on the visible direct radiative forcing. Journal of Quantitative Spectroscopy and Radiative Transfer, 93(4), pp.397-413.

Haywood, J.M. and Ramaswamy, V., 1998. Global sensitivity studies of the direct radiative forcing due to anthropogenic sulfate and black carbon aerosols. Journal of Geophysical Research: Atmospheres, 103(D6), pp.6043-6058.

Guan, H., Schmid, B., Bucholtz, A. and Bergstrom, R., 2010. Sensitivity of shortwave radiative flux density, forcing, and heating rate to the aerosol vertical profile. Journal of Geophysical Research: Atmospheres, 115(D6).

---

## Referee Comment (RC2) · Anonymous Referee #2 · 31 Oct 2019

Review of "Vertical profiles of sub-micron aerosol single scattering albedo over Indian region immediately before monsoon onset and during its development: Research from the SWAAMI field campaign" by Mohanan. R. Manoj et al., (ACP-2019-657)

This paper presents some interesting results on vertical distribution of sub-micron size aerosol characteristics obtained from five different locations over India including Bay-of-Bengal and Arabian Sea. These are obtained from SWAAMI aircraft field campaign

during June-July covering before, during and after the onset of Indian Summer Monsoon. This study claims in obtaining the high resolution vertical profiles of Single Scattering Albedo (SS) and extinction coefficients extending from near surface to about 6 km. Highlight of this study lies in showing an underestimation (over-estimation) of the heating rates over regions with low (high) SSA, when a single SSA value is used. This leads to emphasize the importance of height resolved information particularly in obtaining the heating rates. In addition, significant heating of the atmosphere by sub-micron aerosol absorption in the middle troposphere is reported when realistic profiles are used which is expected to have strong implications on clouds and climate, as authors rightly claimed.

In general, paper is concise and well written with some new information and apt for Atmospheric Chemistry and Physics Journal. However, few clarifications are required before accepting for its publication. Below are the some of the issues which authors need to take care. Authors are strongly encouraged to revise this manuscript.

Major Comments:

1. At several places (for example lines 16-20 in page 12), it is mentioned that the present results matches well with previous reported results (not all but at least few) where they have used single SSA values. Is it not contradicting in saying that height profile of SSA will give more information when compared to use of single value?

2. Page 12, Lines 18-20: 'Strong meridional and zonal gradients in aerosol induced heating rates over AS and BoB across the peninsular landmass (from <0.1 K day-1 over the south - western Arabian Sea, increasing to 0.5 Kday-1 over the north - eastern Bay of Bengal).'

It is mentioned that realistic observations are used in estimating the heating rates. However, it is not mentioned anywhere on what is the uncertainty in estimating the heating rates? Have you considered the uncertainty in several parameters that are used in estimating the heating rates?

3. Page 6, Lines 22-25: It is mentioned that 'Extinction coefficient, decreases near exponentially with altitude over most of the mainland, while over the oceanic regions of the Arabian Sea and Bay of Bengal, an increase in extinction is indicated above 2 km, attributed to elevated layers of aerosols. These layers appear to be stronger over the Arabian Sea than over Bay of Bengal. In general, highest values are observed within 3 km from the surface where the aerosol abundance is more.'

This paragraph has several contradicting statements. Why elevated aerosols are not seen over mainland? In fact several earlier studies have reported elevated aerosol layers over mainland (Mishra et al., 2010; Ganguly et al.,2006; Niranjan et al., 2007; Sinha et al., 2013; Venkat Ratnam et al., 2018). Further, it is mentioned that highest values are observed within 3 km from the surface. Then question arises why they are not washed out after the onset of monsoon?

Minor comments:

1. Page 7, Lines 23-26. 'Examining our values with those reported earlier for this region, based on airborne measurements, Earlier observations over this region have reported a columnar (up to 3 km altitude) mean SSA of 0.86 at 520 nm over Lucknow during pre-monsoon period (Babu et al., 2016) which is just a shade higher than our value of 0.83 ±0.08 (for the altitude range 0-3 km for the same season).'

Previously you have mentioned that SSA has been estimated for 550 nm (Equation 2). Are you comparing the values for same wavelengths?

2. Page 10, Lines 3-5 and also Page 12, Lines 8-9: 'At Lucknow, as stated earlier, significant washout leads to very low extinction above 3 km, while at the lower altitudes, the SSA has increased compared to the pre-monsoon with values remaining around 0.9 up to 2 km and decreasing above to be less than 0.8 at 3 km.'

Why washout should happen only above 3 km. Are you talking about rainout? In fact washout should happen throughout the altitude as mentioned in many recent papers

(for example Venkat Ratnam et al., 2018).

3. There are few grammatical mistakes and typos and hope they are taken care in the editing. However, I suggest authors to go through the manuscript carefully again before submitting revised draft.

Additional references:

Ganguly, D., Jayaraman, A., Gadhavi, H., 2006. Physical and optical properties of aerosols over an urban location in western India: seasonal variabilities. J. Geophys. Res. 111, D24206. http://dx.doi.org/10.1029/2006JD007392.

Mishra, M.K., Rajeev, K., Bijoy, V.T., Parameswaran, K., Nair, A.K.M., 2010. Micro pulse lidar observations of mineral dust layer in the lower troposphere over the southwest coast of Peninsular India during the Asian summer monsoon season. J. Atmos. Sol. Terr. Phys. 72, 1251–1259.

Niranjan, K., Madhavan, B.L., Sreekanth, V., 2007. Micro pulse lidar observation of high altitude aerosol layers at Vishakhapatnam located on the east coast of India. Geophys.Res. Lett. 34, L03815. http://dx.doi.org/10.1029/2006GL0.28199.

Sinha, P.R., Manchanda, R.K., Kaskaoutis, D.G., Kumar, Y.B., Sreenivasan, S., 2013. Seasonal variation of surface and vertical profile of aerosol properties over a tropical urban station Hyderabad, India. J. Geophys. Res. 118. http://dx.doi.org/10.1029/2012JD018039.

Venkat Ratnam, M., P. Prasad, M. Roja Raman, V. Ravikiran, S. Vijaya Bhaskara Rao, B.V. Krishna Murthy, A. Jayaraman, Role of dynamics on the formation and maintenance of the elevated aerosol layer during monsoon season over south-east peninsular India, Atmospheric Environment 188 (2018) 43–49

—END—

---

## Author Comment (AC1) · 11 Dec 2019

At the outset, we thank the reviewer for the observation that "overall, the present work appears to be a scientifically sound exploration of this topic" and also recognizing that "Aerosol-monsoon interactions over South Asia is a very active research topic for which many open questions remain." We fully agree with this. However, while thankfully acknowledging the reviewer's overall appreciation of our paper; we disagree to some of

the comments and inferences, which apparently led to the negative recommendation. Our reasons are given below:

1. The joint INDO-UK campaign SWAAMI was conceived to address to some of the most important gaps in aerosol monsoon interactions; in which, the vertical profiles of SSA over the core Indian monsoon region (that is the Indo-Gangetic Palins, IGP) prior to the onset of Summer monsoon and its maturing phase is identified as one of the key parameters to be measured; as this information is lacking over this region, and this period, when the aerosol system of the IGP is most complex; being a mixture of advected mineral dust and local emissions (including BC, sulphates, nitrates and organics) and the meteorology is in the transformation; with moist air from the northern Indian ocean super posed over the dry air from West Asia and the intense heating of the Gangetic plains (with temperatures typically above 40°C over the entire region, leading to intense convective mixing. The time of the campaign and the type of instruments and the flight paths all were chosen accordingly, and of the several measurements of aerosols, clouds and radiation made using the FAAM aircraft of UK, we address to the specific topic of SSA and its vertical distribution just before and just after the onset of Indian Summer Monsoon to understand how the energetics changes due to change in aerosol distribution and also interaction with the monsoon system. This way, the experiment and its results are scientifically very important.

2. To our knowledge, there are no in situ measurements of SSA over this region, during the monsoon-onset phase, going as high as 6 km or above and in that way our results are the first of their kind and most relevant to the objectives of the bigger campaign. (We address to the specific issues pointed out by the reviewer, separately under other specific comments.) As such, we believe it to be apt for the reputed journal of ACP, which is bringing out a special issue on South Asian Aerosols and monsoon interaction, recognizing its topical importance. As such, we humbly disagree with the reviewers notion implying that this is 'yet another study'. In fact, this is unique, for the first time measurement of SSA over the Indian mainland up to almost mid-troposphere.

3. Further, our results very clearly have brought out the importance of the altitude re-solved SSA and the large difference it makes to the vertical structure of aerosol heating rate over this region than those estimated using the currently available columnar SSA values (derived from sunphotometer networks or combination of space borne data). Such a result is also not reported so far from many places over the globe; but not at all from South Asia.

We have revised the manuscript to bring out the above aspects in focus. Responses to Specific Comments:

1) Significant variation in the SSA was observed during the campaign, with values ranging from near unity to as low as 0.7 in one region".

Yes, we agree and in fact, this itself is a very important finding; especially the low values of SSA at higher altitudes have significant implications for aerosol cloud inter-actions and monsoons, besides in direct atmospheric warming due to absorption of radiation, above low-level monsoon clouds. We have emphasized this aspect in the revised version of the manuscript (page 6, lines 21-22).

2) Only six different profiles are presented, and only one pair of these consists of measurements made before and during the monsoon at the same location.

Yes, we agree and also wish we had more flights from different locations. However, technical and logistical constraints and the delay in getting the aircraft readied before onset of monsoon had put a limitation on the number of sorties that could be made before onset. Nevertheless the results from the limited number of flights are quite significant

3) Many other studies have explored monsoon-aerosol interactions over India using aircraft data (see Li et al., 2016 )-including an analysis of measurements from the same period as the present manuscript (Vaishya et al., 2018)-and it is not clear to me what makes the data presented in the present work particularly unique.

We partly agree and partly disagree. Firstly, there are not 'many other studies' over this region. In fact, there were none before SWAAMI; as has been said in out general response #2. A few aircraft measurements that are available over the IGP pertained to winter and early spring, when the meteorology is different, convection is weak and long-range transport is subdued. Even the paper mentioned by the reviewer (Vaishya et al 2018, which itself was part of the SWAAMI campaign) had limitations in its vertical extent (covering only up to 3 km above ground level) and also pertained only to the scenario prior to onset of monsoon. It did not examine the changes as the monsoon progressed. Our measurements yielded SSA profiles as high as 5 km in its vertical coverage and covered the pre-onset and main phases of the monsoon (page 3, lines 8-13). Presence of absorbing aerosols at higher altitudes (below low-level monsoon clouds) are very important due to two processes; one amplification of absorption due to the reflective clouds underneath and two, the higher warming for same amount of radiation absorbed, due to the thinner air at higher altitudes. These are the uniqueness of our results and we have better focused on these in the revised manuscript (page 10, lines 11-13).

The other measurements available (on which Ramanathan et al 2007 relied upon greatly) are the ones during the Indian Ocean Experiment airborne measurements (Ramanathan et al 2001), and by Corrigan et al 2008; both of which were confined to the south Asian outflow over the Indian Ocean and Southeast Asia, during late winter to early spring when the synoptic meteorology was different. It did not cover any part of the Indian landmass; not even the southern peninsula and thus did not provide the information prior to onset of monsoon, especially over the IGP with its unique characteristics.

On the other hand, the Li et al. (2016) paper is a sort of overview paper, based primarily on modelling and past observational data. The paper did not use any realistic SSA profile over Indian / South Asian landmass (for any season) and this vindicates the absence of this critical information and the dire need for it. This is the knowledge-gap

SWAAMI aimed to bridge. We are revising the ms to bring these novelties to clear focus of the reader.

References

Corrigan, C.E., Roberts, G.C., Ramana, M.V., Kim, D., Ramanathan, V., 2008. Capturing vertical profiles of aerosols and black carbon over the Indian Ocean using autonomous unmanned aerial vehicles. Atmos. Chem. Phys. 8, 737-747.

Li, Z., Lau, W.M., Ramanathan, V., Wu, G., Ding, Y., Manoj, M.G., Liu, J., Qian, Y., Li, J., Zhou, T., 2016. Aerosol and monsoon climate interactions over Asia. Reviews of Geophysics 54, 866-929.

Ramanathan, V., Crutzen, P.J., Lelieveld, J., Althausen, D., Anderson, J., Andreae, M.O., Cantrell, W., Cass, G., Chung, C.E., Clarke, A.D., Collins, W.D., Coakley, J.A., Dulac, F., Heintzenberg, J., Heymsfield, A.J., Holben, B., Hudson, J., Jayaraman, A., Kiehl, J.T., Krishnamurti, T.N., Lubin, D., Mitra, A.P., MacFarquhar, G., Novakov, T., Ogren, J.A., Podgorny, I.A., Prather, K., Prospero, J.M., Priestley, K., Quinn, P.K., Rajeev, K., Rasch, P., Rupert, S., Sadourny, R., Satheesh, S.K., Sheridan, P., Shaw, G.E., Valero, F.P.J., 2001. Indian Ocean Experiment: An integrated analysis of the climate forcing and effects of the great Indo-Asian haze. Journal of Geophysical Research 106, 28,371 - 328,398.

Ramanathan, V., Ramana, M.V., Roberts, G., Kim, D., Corrigan, C., Chung, C., Winker, D., 2007. Warming trends in Asia amplified by brown cloud solar absorption. Nature 448, 575-578.

Vaishya, A., Babu, S.N.S., Jayachandran, V., Gogoi, M.M., Lakshmi, N.B., Moorthy, K.K., Satheesh, S.K., 2018. Large contrast in the vertical distribution of aerosol optical properties and radiative effects across the Indo-Gangetic Plain during the SWAAMI–RAWEX campaign. Atmospheric Chemistry and Physics 18, 17669-17685.

[Figure]

2019.

---

## Author Comment (AC2) · 11 Dec 2019

We appreciate the summary evaluation and the positive recommendation of the reviewer. Our point-by-point responses to the comments of reviewer/ clarification sought are given below and the manuscript is revised accordingly.

Major Comments:

[Figure]

1. At several places (for example lines 16-20 in page 12), it is mentioned that the present results matches well with previous reported results (not all but at least few) where they have used single SSA values. Is it not contradicting in saying that height profiles of SSA will give more information when compared to use of single value?

We are sorry that this paragraph, discussing figure 9 was a bit confusing and thank the reviewer for pointing this out. We have revised the manuscript to avoid confusing statements.

It has been revealed in this study that using height resolved SSA measurements in radiative forcing estimates (rather than a single columnar value) has resulted in the sharp increase in the aerosol induced atmospheric heating around 3 km altitude (Figure 9), rather than the high value near the surface (where aerosol loading and extinction is high) when the single value was used (Figure 9a). In such cases, the heating rates profiles would tend to show higher values where extinction is higher, rather than where SSA is lower, which should be more realistic. This is because of the absence of altitude resolved SSA information, when a single value is used. This under-estimation of heating rate while using a single SSA value becomes severe when the elevated aerosol layers have more absorbing aerosols leading to lower SSA than those closer to the surface as can be seen in Figure 9c. In addition to the effect due to lower SSA, elevated aerosol layers will tend to increase the heating rate at higher altitudes, because for the same amount of radiation absorbed, more heating would be produced at higher regions of the atmosphere, where the air is rarer. Consequence of this is clearly seen in Figure 9d, where the heating rate profiles using single SSA and using height resolved SSA are quite similar, yet there is significant underestimation in the instantaneous atmospheric forcing by as much as ∼2.4 W/m2. In some of the previous studies, which have reported peaks in the heating rate profiles around 2-4 km altitude region, would have this under-estimation, as those did not use height resolved SSA information. With the use of height resolved SSA, our study provides a more realistic estimate of the heating; with higher values around more absorbing aerosol layers.

In Figure 9b the SSA values vary between 0.95 and 0.87. To clearly demonstrate the effect of altitude resolved SSA on heating rate profiles, we have shown the heating rate profiles below (Fig. 1) in which we have plotted the heating rate profiles calculated for the same extinction profile, but with two single SSA values at the extremes (0.87 and 0.95) and also the realistic altitude varying SSA. Though the position of the peak occurs around the same altitude, the peak heating rate as well as integrated heating rates differ significantly. The importance of our study is to demonstrate this through measurements. This is now clearly provided in the revised manuscript (page 12, line numbers 10 and 12-15).

2. Page 12, Lines 18-20: 'Strong meridional and zonal gradients in aerosol induced heating rates over AS and BoB across the peninsular landmass (from <0.1 K/day over the south - western Arabian Sea, increasing to 0.5 K/day over the north - eastern Bay of Bengal).'

Here we have referred to a result reported earlier by Nair et al., 2013. This study, though used altitude resolved temperature, pressure and humidity in RF estimation, did use only columnar value of SSA, because the height resolved SSA was not available. This was recognized as a major gap area and SWAAMI made special efforts to close this gap as much as possible. Our measurements providing height-resolved SSA up to 6 km is a major contribution towards this. This is made explicit in the revised manuscript (page 12, line numbers 27-29).

2a. It is mentioned that realistic observations are used in estimating the heating rates. However, it is not mentioned anywhere on what is the uncertainty in estimating the heating rates? Have you considered the uncertainty in several parameters that are used in estimating the heating rates?

Yes, we have considered the uncertainties due to the parameters used to estimate the forcing. The sensitivity of various parameters including Aerosol Optical Depth (AOD), SSA and asymmetry parameter has been estimated following McComiskey et

al. (2008). The uncertainty due to the combined effect of AOD, SSA and asymmetry parameter on the top of the atmosphere forcing was found to be $\sim$ 1.8 W/m2. Estimation of uncertainty has been discussed in detail in many papers from the group (Moorthy et al., 2009, Satheesh et al., 2010) and hence not repeated here. We also found that the uncertainty in the instantaneous atmospheric forcing for the cases shown in figure 9 was $\sim$4.2 W/m2. Unfortunately, the value was not explicitly mentioned in the manuscript. The magnitudes of the atmospheric forcing estimated with and without the use of SSA profiles are shown in Figure 9. The atmospheric forcing estimates increased by $\sim$14.6% (8.7% to 24.09%) on an average, on using the measured SSA profiles. This is made explicit in the revised manuscript (page 10, line numbers 28-29 and page 11, line numbers 1-2).

3. Page 6, Lines 22-25: It is mentioned that 'Extinction coefficient', decreases near exponentially with altitude over most of the mainland, while over the oceanic regions of the Arabian Sea and Bay of Bengal, an increase in extinction is indicated above 2 km, attributed to elevated layers of aerosols. These layers appear to be stronger over the Arabian Sea than over Bay of Bengal. In general, highest values are observed within 3 km from the surface where the aerosol abundance is more.' This paragraph has several contradicting statements. Why elevated aerosols are not seen over mainland? In fact several earlier studies have reported elevated aerosol layers over mainland (Mishra et al., 2010; Ganguly et al.,2006; Niranjan et al., 2007; Sinha et al., 2013; Venkat Ratnam et al., 2018). Further, it is mentioned that highest values are observed within 3 km from the surface. Then question arises why they are not washed out after the onset of monsoon?

Thank you for pointing out this, we have revised the sentences to avoid confusion and to make the point clear (page 6, line numbers 23-24). The variation in the extinction coefficient profiles is much sharper over the land compared to the oceanic region. At an altitude of 3 km over the land, the extinction coefficient values fall to less than 50% of the value near the surface. Elevated aerosol layers are observed all over the

Indian mainland and the surrounding oceans during pre-mosoon and early monsoon season, due to lofted dust (regional as well as advected). Over the oceanic region the magnitudes of extinction coefficients as well as the fall in the magnitudes of extinction coefficients with height are lower compared to that over the mainland. Over the Arabian Sea and Bay of Bengal elevated aerosol layers are observed above 2 km. The magnitude of extinction coefficient in these layers are larger over the Arabian Sea than over the Bay of Bengal, because of the proximity of Arabian Sea to major dust sources of west Asia and western India. In general, the strongest elevated layers appear within 3 km from the surface.

Minor comments:

1. Page 7, Lines 23-26. 'Examining our values with those reported earlier for this region, based on airborne measurements, Earlier observations over this region have reported a columnar (up to 3 km altitude) mean SSA of 0.86 at 520 nm over Lucknow during pre-monsoon period (Babu et al., 2016) which is just a shade higher than our value of 0.83 ±0.08 (for the altitude range 0-3 km for the same season).' Previously you have mentioned that SSA has been estimated for 550 nm (Equation 2). Are you comparing the values for same wavelengths?

No, we are not comparing the SSA here as the values are reported for different wavelengths. Although SSA has a wavelength dependence, the magnitude of SSA is almost the same in 520-550 nm range. Mishra et al. (2015) have shown that the SSA has very small variation (approximately ±0.02) in this wavelength region for dust, polluted dust and pollution aerosols. We want to show that low SSA values similar to our observations have been reported earlier from the region in the 0-3 km altitude range, for the same season. We have modified the statement to convey this clearly (page 7, line numbers 25-26).

2. Page 10, Lines 3-5 and also Page 12, Lines 8-9: 'At Lucknow, as stated earlier, sig-nificant washout leads to very low extinction above 3 km, while at the lower altitudes,

the SSA has increased compared to the pre-monsoon with values remaining around 0.9 up to 2 km and decreasing above to be less than 0.8 at 3 km.' Why washout should happen only above 3 km. Are you talking about rainout? In fact washout should happen throughout the altitude as mentioned in many recent papers (for example Venkat Ratnam et al., 2018).

Yes, it is mostly washout. In the IGP region (near Lucknow) high altitude aerosols mainly consist of transported dust, while near the surface it is a mixture of transported and locally produced aerosols. The effect of rain is seen throughout the column. The magnitude of extinction coefficients reduces even at lower altitudes during monsoon. The aerosols removed from near the surface are quickly replenished, due to large local production. The aerosols are not replenished as quickly at higher altitudes. We agree that the sentence gives the wrong impression that the removal of aerosols is happening only above 3 km. We have modified the statement to convey this clearly (page 10, line number 4 and page 12, line number 10). Our observations cannot be completely explained by the cases discussed in Ratnam et al., (2018) as 1) our measurements do not show the extinction profiles immediately after rain and 2) the measurements were made around the local noon when the local sources are active.

References:

Mishra, A.K., Koren, I., Rudich, Y., 2015. Effect of aerosol vertical distribution on aerosol-radiation interaction: A theoretical prospect. Heliyon 1.

McComiskey, A., Schwartz, S.E., Schmid, B., Guan, H., Lewis, E.R., Ricchiazzi, P., Ogren, J.A., 2008. Direct aerosol forcing: Calculation from observables and sensitivities to inputs. Journal of Geophysical Research: Atmospheres 113.

Moorthy, K. K., Nair, V. S., Babu, S. S., and Satheesh, S. K.: Spatial and vertical heterogeneities in aerosol properties over oceanic regions around India: Implications for radiative forcing, Quarterly Journal of the Royal Meteorological Society, 135, 2131-2145, 10.1002/qj.525, 2009.

Nair, V.S., Babu, S.S., Moorthy, K.K., Prijith, S.S., 2013. Spatial Gradients in Aerosol-Induced Atmospheric Heating and Surface Dimming over the Oceanic Regions around India: Anthropogenic or Natural? Journal of Climate 26, 7611-7621.

Ratnam, M.V., Prasad, P., Roja Raman, M., Ravikiran, V., Bhaskara Rao, S.V., Krishna Murthy, B.V., Jayaraman, A., 2018. Role of dynamics on the formation and maintenance of the elevated aerosol layer during monsoon season over south-east peninsular India. Atmospheric Environment 188, 43-49.

Satheesh, S. K., Vinoj, V., and Krishnamoorthy, K.: Assessment of aerosol radiative impact over oceanic regions adjacent to Indian subcontinent using multisatellite analysis, Advances in Meteorology, 2010, 2010.

──────────────────────────────

[Figure]

[Figure]

**Fig. 1.** Fig. 1: The heating rate profile shown in Figure 9b in the manuscript is reproduced along with the profiles corresponding to single SSA values 0.87 and 0.95.